# Chitosan Nanocomplexes for the Delivery of ENaC Antisense Oligonucleotides to Airway Epithelial Cells

**DOI:** 10.3390/biom10040553

**Published:** 2020-04-05

**Authors:** A. Katharina Kolonko, Nadine Bangel-Ruland, Francisco M. Goycoolea, Wolf-Michael Weber

**Affiliations:** 1Institute of Animal Physiology, University of Muenster, Schlossplatz 8, 48143 Muenster, Germany; n.br@wwu.de (N.B.-R.); wmw@wwu.de (W.-M.W.); 2School of Food Science and Nutrition, University of Leeds, Leeds LS2 9JT, UK; F.M.Goycoolea@leeds.ac.uk

**Keywords:** CFTR, ENaC, chitosan, antisense oligonucleotides, nanocomplexes, cystic fibrosis, nanomedicine, drug delivery

## Abstract

Nanoscale drug delivery systems exhibit a broad range of applications and promising treatment possibilities for various medical conditions. Nanomedicine is of great interest, particularly for rare diseases still lacking a curative treatment such as cystic fibrosis (CF). CF is defined by a lack of Cl^−^ secretion through the cystic fibrosis transmembrane conductance regulator (CFTR) and an increased Na^+^ absorption mediated by the epithelial sodium channel (ENaC). The imbalanced ion and water transport leads to pathological changes in many organs, particularly in the lung. We developed a non-viral delivery system based on the natural aminopolysaccharide chitosan (CS) for the transport of antisense oligonucleotides (ASO) against ENaC to specifically address Na^+^ hyperabsorption. CS–ASO electrostatic self-assembled nanocomplexes were formed at varying positive/negative (P/N) charge ratios and characterized for their physicochemical properties. Most promising nanocomplexes (P/N 90) displayed an average size of ~150 nm and a zeta potential of ~+30 mV. Successful uptake of the nanocomplexes by the human airway epithelial cell line NCI-H441 was confirmed by fluorescence microscopy. Functional Ussing chamber measurements of transfected NCI-H441 cells showed significantly decreased Na^+^ currents, indicating successful downregulation of ENaC. The results obtained confirm the promising characteristics of CS as a non-viral and non-toxic delivery system and demonstrate the encouraging possibility to target ENaC with ASOs to treat abnormal ion transport in CF.

## 1. Introduction

There is increasing interest in the development of non-viral gene delivery systems to overcome viral safety concerns such as immunogenicity and insertional mutagenesis. Amongst others, these delivery systems should fulfil a variety of requirements such as low cytotoxicity, applicable flexibility and low manufacturing costs. Non-viral vectors are roughly class-divided into cationic lipids and cationic polymers [1]. Although synthetic cationic lipids show very good transfection efficiency and are commonly used, these nanocomplexes are known to be more or less cytotoxic, drawing the attention of this study to natural cationic polymers. There are several diseases that would benefit from an efficient therapeutic delivery system to transport a variety of compounds to certain tissues. One of these diseases currently lacking a curing treatment is cystic fibrosis (CF).

CF is a chronic, life-shortening and rare hereditary disease, causing severe damage to the respiratory tract and other organs. One affected newborn in 3000 births makes the disease the most common autosomal recessive inherited disorder in the Caucasian population, with a total of approximately 70,000 patients worldwide [2]. The monogenetic disease is caused by mutations in the gene coding for cystic fibrosis transmembrane conductance regulator (CFTR)*,* a cAMP-dependent chloride and bicarbonate channel located in the apical membranes of secretory epithelial cells [3]. The loss of function leads to pathological changes in organs that express the chloride channel such as lung, pancreas, intestine, liver and reproductive tract, evoking a wide range of symptoms [4]. 

In addition to its function as a chloride channel, CFTR is a central regulatory protein, with extensive regulation influence on different epithelial transport systems [5]. The second major abnormality in CF airway epithelia after the loss of Cl^−^ secretion is a dramatic increase in Na^+^ absorption caused by an increased apical Na^+^ permeability through the epithelial sodium channel (ENaC). As a result, the rate of transepithelial Na^+^ absorption in vitro and in vivo is 2- to 3-fold larger in CF than in normal epithelia [6,7]. The concomitant dysregulation of the two ion channels leads to an imbalance of ion and water transport in epithelial cells, causing severe problems, particularly in the lung. The excess of Na^+^ absorption entails an increased influx of water, leading to dehydration and depletion of the airway surface liquid (ASL) covering the epithelial cells in the respiratory tract. The loss of Cl^−^ secretion and water efflux prevents the correction of the low ASL volume. As a result, the reduced ASL and hyperconcentrated mucus weigh down the cilia, impairing ciliary beating and hence the normal mucociliary clearance, thus causing CF lung disease [8]. CF lung disease is defined by chronic airway infection, progressing to bronchiectasis, gas trapping, hypoxemia and hypercarbia. The resulting pulmonary insufficiency is responsible for approximately 80% of CF-related deaths [9].

Previously, we developed an innovative strategy to address the underlying defect using potent DNA-based technologies. We successfully demonstrated that ENaC-specific antisense oligonucleotides (ASO) could evoke a long-lasting repression of the excessive Na^+^ absorption in CF [10]. These molecules in general have become a valuable tool in many different studies and clinical trials inhibiting gene expression [11]. An ongoing clinical trial is currently testing the drug eluforsen, an ASO designed to restore CFTR protein function in the airway epithelium, demonstrating the high significance of these molecules for CF research [12].

To provide an efficient and safe respiratory absorption of these therapeutically efficacious compounds, we are currently working on a gene delivery system based on chitosan (CS) to specifically address the requirements of CF. CS, the main derivate of chitin, refers to a family of pseudonatural linear polysaccharides composed of randomly distributed β-(1-4)-linked D-glucosamine and *N*-acetyl-D-glucosamine units [13]. The polycationic character of CS enables it to interact with and bind to negatively charged nucleic acids, mucins and sulphated glycosylaminoglycans in the glycocalyx and other drugs. This property, along with its biocompatibility, mucoadhesiveness and low cytotoxicity [14,15], as well as biodegradability in humans by lysozyme and several chitinases [16], makes CS a very promising candidate for non-viral gene delivery. So far, it has been used for a variety of oral and nasal drug delivery applications [17,18]. In the past, CS was already used in a variety of studies to deliver ASOs to target cells [19,20,21,22]. However, to the best of our knowledge, we provide the first attempt at delivering ASOs using CS to address the imbalanced ion transport in CF. Previously, we showed that transfection of nucleic acids using CS is an effective delivery system in a human CF cell line [23,24]. Now, we are interested in gleaning understanding of whether this approach is adaptable to an air–liquid interface (ALI) culture system, which is characterized by an extensive mucociliary differentiation process resulting in an in vitro model that is representative of the in vivo airway physiological context.

We formed CS–ASO nanocomplexes by electrostatic self-assembly at different positive/negative (P/N) charge ratios using CS working solutions with or without NaCl and two different ASOs against the α-subunit of ENaC as well as an ASO sense control. The nanocomplexes were then characterized in terms of their physicochemical properties such as size, the polydispersity index (PDI), and zeta potential, as well as their binding efficiency and stability. After choosing the most suitable formulation, we demonstrated that these nanocomplexes have a very low cytotoxic effect on human epithelial cells compared to conventional transfection particles. Furthermore, we verified successful cellular uptake of CS–ASO nanocomplexes by using fluorescently labelled ASOs. Finally, we evaluated the efficiency of CS–ASO nanocomplexes by performing electrophysiological Ussing chamber measurements, demonstrating successfully decreased Na^+^ currents mediated by ENaC.

## 2. Materials and Methods 

### 2.1. Preparation of Nanocomplexes

The CS used was an ultrapure biomedical-grade sample (Heppe 70/5; Batch-No. 212-170614-01; DA = 17%; Mw = 29.3%) purchased from HMC+ GmbH (Halle/Saale, Germany). It was dissolved in water with or without 85 mM NaCl, both added with 5% stoichiometric excess of 5 M HCl. The CS stock solutions were diluted to reach the desired concentrations for varying P/N charge ratio (defined as the molar ratio of amine to phosphate groups) working solutions. 

In this study, the already established ASO against the α-subunit of ENaC (accession number: NM_001038) was used [10]. The ASO sequence termed “ASOgreen” was designed by Segal et al. and corresponds to the human α-ENaC mRNA at position 290 [10,25]. For control experiments, a sense control strand and an ASO labelled with a fluorescent 6-Fam tag were used (Table 1). The nucleotides were synthesized as phosphorothioates by metabion international AG (Planegg, Germany).

CS–ASO nanocomplexes of different P/N charge ratios were prepared by mixing a constant amount of an ASO with 2 × volumes of the CS working solutions (Table 2). Briefly, 20 µL CS working solution with or without 85 mM NaCl was mixed with 10 µL ASO and 20 µL water and incubated for 30 min at room temperature (RT) to allow the spontaneous self-assembly of the nanocomplexes.

### 2.2. Determination of Size Distribution and Zeta Potential

The size distribution of the nanosystems was measured by dynamic light scattering with non-invasive back scattering (DLS-NIBS) with a measurement angle of 173°. The nanosystems were diluted 1:7 in water. Zeta potential was acquired by applying the Henry equation using the Smolouchowski approximation after exerting laser Doppler microelectrophoresis and phase analysis light scattering. The nanosystems were diluted 1:33 in 1 mM KCl. All measurements were conducted with a Zetasizer Nano ZS 6300 (Malvern Panalytical Ltd., Worcestershire, UK) instrument using a folded capillary zeta potential cell (Model DTS1070; Malvern Panalytical Ltd., Worcestershire, UK) at 25 °C.

### 2.3. Stability Measurements

The stability of the nanosystems in the transfection medium was determined by measuring their size distribution at different time points by DLS-NIBS with a measurement angle of 173° using a Zetasizer Nano ZS 6300 (Malvern Panalytical Ltd., Worcestershire, UK) instrument. Measurements were conducted at 37 °C using a low volume disposable cuvette (Sarstedt AG & Co, Hemer, Germany). The nanosystems were diluted 1:20 in Opti-MEM™ (Thermo Fisher Scientific, Waltham, MA, USA) or Opti-MEM™ supplemented with HEPES (20 mM) and mannitol (270 mM) and incubated at 37 °C in between measurements.

### 2.4. Gel Retardation Assay

To test the binding efficiency of nucleic acids to CS, a gel retardation assay was carried out. Briefly, nanocomplexes were prepared at different P/N charge ratios as described above and loaded onto a 1.5% agarose gel in 0.5× TBE buffer supplemented with 0.006% Midori Green Advance (Nippon Genetics Europe GmbH, Dueren, Germany) and electrophoresed at 128 V for 40 min. Finally, the nucleic acid bands were visualized in a BioDocAnalyze System (Analytik Jena, Jena, Germany).

### 2.5. Cell Culture

The cell line NCI-H441 (ATCC^®^ HTB174™) was a kind gift from Prof. Dr. Mike Althaus (Bonn-Rhein-Sieg University of Applied Sciences, Rheinbach, Germany). NCI-H441 cells were cultured in uncoated T75 flasks in RPMI 1640 medium with supplements (1% antibiotic/antimycotic solution, 1% sodium pyruvate, 1% ITS media supplement, and 10% fetal bovine serum) at 37 °C, 5% CO_2_ and 95% rH. For Ussing chamber experiments, cells were seeded onto uncoated Costar Transwell^®^permeable filters (Ø = 6.5 mm; REF 3470; Corning Inc., Lowell, MA, USA) with a density of 1 × 10^6^ cells/cm^2^. One day after seeding, the NCI-H441 medium was replaced by NCI-H441 medium supplemented with 200 nM dexamethasone. For cultivation of the cells under ALI conditions, the medium on the apical side of the filters was removed one day after seeding and cells were only supplied with the medium on the basolateral side. The filters were cultured for at least 7 days at 37 °C, 5% CO_2_ and 95% rH before experiments were conducted. For transfection experiments with NCI-H441 cells grown on filters, the medium was replaced by antibiotic-free NCI-H441 medium without dexamethasone 24 h before transfection. For fluorescence optical experiments, cells were seeded on glass cover slips (Ø = 12 mm; Carl Roth GmbH & Co. KG, Karlsruhe, Germany) with a density of 1 × 10^5^ one day before transfection and incubated in antibiotic-free medium.

### 2.6. MTT Assay

Cytotoxicity of the nanoformulations and components was tested using a 3-(4,5-dimethylthiazol-2-yl)-2,5-diphenyltetrazolium bromide (MTT) assay. Briefly, cells were seeded in a 96-well microtiter plate with a density of 1 × 10^4^ cells per well and incubated at 37 °C, 5% CO_2_ and 95% rH for 24 h. The cells were washed twice with serum-free cell culture medium before the samples were added and the cells were incubated under previous conditions for 24 h. After incubation, the samples were removed and replaced by 100 µL serum-free cell culture medium and 25 µL MTT solution (5 mg/mL in PBS). The cells were incubated under previous conditions for another 4 h to allow the formation of a purple formazan salt before the medium was removed and 100 µL dimethyl sulfoxide was added to each well in order to dissolve the formazan. After the well plates were incubated at 37 °C for 30 min, the absorbance was measured at λ = 570 nm in a microplate reader (EZ Read 400, Biochrom GmbH, Berlin, Germany). Relative viability was calculated by dividing individual viabilities by the mean of the negative control (serum-free cell culture medium). The 1% Triton^®^ X-100 was used as a positive control.

### 2.7. Transfection

Twenty-four hours before transfection, the medium of the cells was replaced with fresh antibiotic-free medium. CS–ASO nanocomplexes were prepared with the desired amount of nucleic acid in 85 mM NaCl to reach a P/N charge ratio of 90 as described above. The formulations were incubated at RT for 30 min to allow the self-assembly of the nanocomplexes. After the incubation, the nanocomplexes were mixed with Opti-MEM™ to reach a final volume of 300 µL (filter) or 500 µL (glass cover slip) and incubated for another 5 min at RT. Finally, the nanocomplexes were added to the cells and cells were incubated for 24 h at 37 °C, 5% CO_2_ and 95% rH before experiments were conducted. As a control, cells were transfected using the commercially available transfection reagent Lipofectamine^®^2000 Reagent (Lipofectamine; Invitrogen, Karlsruhe, Germany). Transfection was conducted according to the manufacturer’s instructions using 2 µL per Transwell^®^permeable filter or glass cover slip.

### 2.8. Fluorescence Optical Experiments

In order to verify the success of transfection procedures, cells seeded on glass cover slips were transfected with 5′Fam-ASOgreen containing the fluorescent 6-Fam tag (0.15 µg per glass cover slip) and incubated for 24 h at 37 °C with 5% CO_2_ and 95% rH. As a control, cells were transfected with the non-fluorescent ASOgreen. Transfected cells on glass cover slips were washed three times with PBS. Subsequently, cells were fixed with 500 µL 3.5% paraformaldehyde (PFA) in PBS for 30 min. After two more washing steps with PBS, cells were incubated for 10 min in 500 µL glycine (100 mM in PBS) in order to quench the residual PFA. Cells were washed three more times with PBS and finally placed top down onto microscope slides on a drop of mounting medium Fluoroshield™ with DAPI (Sigma-Aldrich, St. Louis, MO, USA). Fixed cells were analyzed using the confocal laser scanning microscope (CLSM) LSM 510 META (Carl Zeiss AG, Oberkochen, Germany) and the program LSM 5 (Carl Zeiss AG, Oberkochen, Germany). Analysis of total fluorescence intensities was carried out using the plug in for RGB (red, green, blue) intensity measurement in the program ImageJ (Version 1.48v, Wayne Rasband, National Institutes of Health, Bethesda, MD, USA).

### 2.9. Transepithelial Measurements

Transepithelial measurements on NCI-H441 cells were performed in modified Ussing chambers designed by Prof. Dr. Willy Van Driessche (EP-Devices, Leuven, Belgium). We used Ag/AgCl electrodes, which were connected to the Ringer solution. The V_t_ was clamped to 0 mV with a low-noise voltage clamp. After adapting the cells to the Ringer solution (130 mM NaCl, 5 mM KCl, 2 mM MgCl_2_, 1 mM CaCl_2_, 5 mM glucose, 10 mM HEPES; pH 7.3; 37 °C), Na^+^ absorption through ENaC was blocked by apical application of amiloride (10 µM in Ringer solution). To determine the overall Na^+^ absorption of the cells, Na^+^-free Ringer solution (130 mM tetramethylammonium chloride instead of NaCl) was applied to the apical side in a second step. The transepithelial short-circuit current (I_sc_) was continuously recorded (ImpDsp 1.4; KU Leuven, Leuven, Belgium). All values were normalized to 1 cm^2^.

### 2.10. Statistical Analysis

The statistical analysis was performed with GraphPad Prism^®^ Version 6.01 (GraphPad Software Inc., La Jolla, CA, USA). For analysis of the data, the arithmetic mean values ± standard deviations (SD) of at least three independent experiments were determined, if not stated otherwise. Ussing chamber data are expressed as the arithmetic mean values ± standard error of the mean (SEM). In order to assess the significant differences of the results, the non-parametric Kruskal–Wallis test was performed. Differences were considered statistically significant when *p* ≤ 0.05 (*), *p* ≤ 0.01 (**), *p* ≤ 0.001 (***), and *p* ≤ 0.0001 (****).

## 3. Results

### 3.1. Physicochemical Characterization of Nanocomplexes

A complete characterization of the physicochemical properties was performed for all nanocomplexes individually. The Z-average hydrodynamic diameter and the polydispersity index (PDI) of the nanocomplexes were determined by DLS-NIBS. The particle size of all nanocomplexes varied between 100 and 200 nm (Figure 1). Not much variation was observed between complexes of CS–ASOgreen and CS–5′Fam-ASOgreen of varying P/N charge ratio nor between water and NaCl (Figure 1a,c). However, the CS–ASOgreen_sense systems made in 85 mM NaCl did show a notorious increasing trend in size with P/N charge ratio (Figure 1b). PDI measurement revealed that on average nanocomplexes prepared in the presence of NaCl had a somewhat lower PDI than systems prepared without the salt. Furthermore, a decrease in PDI with increasing P/N charge ratio could be observed with the lowest PDI in nanocomplexes at a P/N charge ratio of 90 (Figure 2a,b). Overall, the PDI varied between 0.15 and 0.35, indicating that the nanocomplexes are in general monodisperse. The results obtained from DLS-NIBS suggest that the nanocomplexes prepared with NaCl are in general slightly more monodisperse than those prepared in water, in correspondence with the PDI measurements.

The zeta potential of the nanocomplexes was determined from their electrophoretic mobility and is displayed in Figure 3. The zeta potential increased with increasing P/N charge ratio from approximately +25 to +35 mV. On average nanocomplexes prepared in the presence of 85 mM NaCl had a lower zeta potential than nanocomplexes prepared without NaCl. Of note, the non-linear dependence of zeta potential with P/N ratio observed between the CS–ASOgreen and CS–ASOgreen_sense is different (Figure 3a,b, respectively), with a greater exponent for the latter. In addition, the nanocomplexes formed with the fluorescently labelled ASO showed a linear dependence between zeta potential and P/N ratio by contrast with the other two systems that showed a non-linear profile.

To evaluate the binding efficiency of CS and ASOs, a gel retardation assay was performed (Figure 4). Due to its negative charge, the naked ASO ran through the gel and showed a specific band below 100 bp corresponding to their size of 16 bp. Positively charged CS was retained in the pocket. CS–ASO nanocomplexes at different P/N charge ratios did not show a specific band as the negatively charged nucleic acid was retained in the pocket by the formed nanocomplexes, whereas the retained amount of ASOs appeared to be proportional to the P/N charge ratio.

After careful consideration of the assessed physicochemical properties, nanocomplexes at a P/N charge ratio of 90 prepared in the presence of 85 mM NaCl were chosen as the prototype formulations for transfection experiments. Table 3 summarizes the Z-average hydrodynamic diameter, PDI and zeta potential of the chosen nanocomplexes.

### 3.2. Stability of CS–ASO Nanocomplexes

The stability of the nanocomplexes in the cell culture transfection medium was determined by measuring their size by DLS-NIBS during incubation. Nanocomplexes were incubated in Opti-MEM™ or Opti-MEM™ supplemented with HEPES and mannitol at 37 °C for 24 h. Figure 5 shows the Z-average hydrodynamic diameter of the nanocomplexes in the media. CS–ASOgreen nanocomplexes in Opti-MEM™ supplemented with HEPES and mannitol were stable for the first hour of incubation, while nanocomplexes in Opti-MEM™ without supplements aggregated immediately. However, after approximately 5 h, the systems settled at approximately 2000 nm. The other two systems seemed to be less stable in supplemented Opti-MEM™ during the first five hours. However, they also settled at longer incubation periods. Overall, nanocomplexes appeared to be slightly more stable in Opti-MEM™ supplemented with HEPES and mannitol. Visible aggregation was not observed in either medium at any time.

When working with cells in culture, it is crucial to maintain optimal conditions such as temperature or CO_2_ concentration to minimize cell stress. Therefore, the osmolality of the transfection media was determined by a cryoscopic osmometer. Table 4 displays the osmolality of NCI-H441 cell culture medium as well as the two different transfection media. While the osmolality of Opti-MEM™ without supplements was similar to that of the NCI-H441 cell culture medium, the osmolality of Opti-MEM™ supplemented with HEPES and mannitol was more than twice as high. This high osmolality could severely affect cellular homeostasis and impair the transfection. Therefore, transfection experiments were conducted with Opti-MEM™ without supplements, even though nanocomplexes were more stable in the medium supplemented with HEPES and mannitol.

### 3.3. Cell Culture Experiments with CS–ASO Nanocomplexes

Before conducting transfection experiments with CS–ASO nanocomplexes, the effect of the nanocomplexes on cell viability was determined. NCI-H441 cells were incubated with the nanocomplexes and controls for 24 h. Subsequently, a MTT assay was carried out. The test showed that CS–ASO nanocomplexes did not have a significant effect on the viability of the cells, while Lipofectamine-containing systems displayed highly significant cytotoxicity (Figure 6). Furthermore, the test confirmed the negative impact of the high osmolality of Opti-MEM™ supplemented with HEPES and mannitol as the transfection medium displayed a highly significant impairment of the cell viability compared to Opti-MEM™ without supplements. After ruling out cytotoxicity of the nanocomplexes, transfection experiments were performed.

Transfection experiments were performed with NCI-H441 cells using the fluorescently labeled ASO in order to verify successful cellular uptake. Cells were seeded on glass cover slips and transfected with 5′Fam-ASOgreen using Lipofectamine or CS. As a control, cells were transfected with the non-fluorescent ASOgreen. Figure 7 depicts representative CLSM images of the transfected cells. While the ASOs transfected with Lipofectamine appear very bright and clear, the nucleotides transfected with CS seem more diffuse and dimmed. However, a quantitative comparison of total RGB fluorescent intensities revealed no significant difference between transfection performed with Lipofectamine and CS (Figure 8). In fact, the significant difference between control cells and cells transfected with the fluorescently labeled ASOs was higher when using CS as transfection reagent as compared to Lipofectamine. Having proven successful cellular uptake of CS–ASO nanocomplexes, further transfection experiments were conducted.

In order to test the ability of the ASOs to functionally downregulate ENaC activity, NCI-H441 cells were transfected with 0.45 µg/cm^2^ ASOgreen and ASOgreen_sense as controls using CS and Lipofectamine, respectively. Twenty-four hours after transfection, Ussing chamber measurements were conducted to monitor ENaC function. During the measurements, ENaC was blocked by the specific blocker amiloride followed by complete apical withdrawal of Na^+^ with the purpose of determining the amount of amiloride-sensitive Na^+^ current. In non-transfected control cells, only a very slight decrease in short-circuit current was observed after the removal of Na^+^ (Figure 9a). This suggests that most of the Na^+^ current is mediated by ENaC. On the contrary, in cells transfected with ASOgreen using CS, the decrease in short-circuit current after application of Na^+^-free Ringer solution was much greater, indicating that only a minimal amount of Na^+^ current is mediated by ENaC (Figure 9b). Statistical evaluation of the short-circuit current confirmed these observations (Figure 9c). After transfection with ASOgreen, the amiloride-sensitive current decreased notably compared to non-transfected control cells. In cells transfected using CS, the decrease was significant (48.4 ± 8.5% vs. 8.9 ± 2.4%; *p* ≤ 0.01), thus demonstrating successful downregulation of ENaC by ASOgreen. Of note, CS nanocomplexes outperformed Lipofectamine transfections. Transfection with the sense control ASOgreen_sense led to a slight decrease in amiloride-sensitive current. However, no significant difference to control cells was identified, confirming, as expected, that ASOgreen_sense had no effect on ENaC expression.

## 4. Discussion

### 4.1. Salt Benefits the Formation of CS–ASO Nanocomplexes

In order to optimize the function of nanocomplexes as a transfection reagent, it is crucial to characterize them for their physicochemical properties and understand them on a molecular level. CS–ASO nanocomplexes were formed at varying P/N charge ratios (30, 50, 70, 90, 100) in either water or in 85 mM NaCl according to the principle of electrostatic self-assembly by spontaneously mixing the components in aqueous solution. Electrostatic self-assembly is defined as the assembly of molecules that are connected by non-covalent interactions, forming via the association of small building blocks [26]. Three different nanocomplexes were designed, incorporating an ASO directed against the α-subunit of ENaC, a sense control and a fluorescently labeled ASO, respectively. Finally, physicochemical properties of the particles were determined by DLS-NIBS and zeta potential measurements.

The hydrodynamic diameter of the nanocomplexes varied between ~100 and ~200 nm (Figure 1) while no apparent trend regarding the P/N charge ratio was observed. However, on average, nanocomplexes formed in the presence of NaCl appeared to be of a smaller size. Higher ionic strength has been shown to be beneficial for CS particle formation, decreasing the particle size with increasing NaCl concentrations [27,28]. NaCl screens out the charges of CS and thereby lowers repulsive electrostatic forces, potentially reducing aggregation. Furthermore, NaCl reduces the stiffness of CS chains, leading to the formation of more compact particles. This is reflected in the measurements of the PDI of the nanocomplexes, as nanocomplexes formed in the presence of NaCl displayed a lower PDI than nanocomplexes formed without salt (Figure 2). Similar results have been reported by Sawtarie et al., who observed a narrower size distribution of CS–tripolyphosphate particles with increasing NaCl concentration, reinforcing the theory [29]. Interestingly, the nanocomplexes of CS–ASOgreen_sense in 85 mM NaCl, showed a consistent increase in size with P/N charge ratio. This result evidenced the possibility that CS and ASOgreen_sense formed nanocomplexes different in kind to those of ASOgreen, a suggestion that was also confirmed by the results of the of zeta potential as explained below (Figure 3).

Measurements of the electrophoretic mobility of nanocomplexes revealed increasing zeta potential, from approximately +25 to +35 mV (Figure 3). The increase in zeta potential with increasing P/N charge ratio indicates that nanocomplexes were successfully formed, as the positive charge arises from the protonated amino groups that were not neutralized by the negatively charged phosphate groups of the ASO. With increasing P/N charge ratio, the amount of CS is increased and therefore an increasing excess of positively charged amino groups leads to an increasing zeta potential as shown in previous studies [23,30]. Furthermore, it was observed that nanocomplexes prepared with NaCl had a slightly lower zeta potential. This can be explained by the ability of the salt to screen charges and coincides with the findings of other groups [27,31]. All in all, the positive zeta potential of approximately +30 mV is an indication for a stable formulation of nanoparticles, as a minimum of +30 mV is required to stabilize nanocomplexes solely by electrostatic repulsion [32]. Yet another unexpected result was to observe differences in the zeta potential dependence on P/N ratio between CS–ASOgreen and CS–ASOgreen_sense nanocomplexes (Figure 3a,b). This unexpected result hints to differences in the actual molecular architecture of the formed nanocomplexes. So far, CS–gene nanocomplexes are assumed to be formed by non-specific electrostatic self-assembly. However, it is possible that specific sequences of CS given by the pattern of acetylation display high specific binding affinity for given nucleotide sequences. To test the validity of this hypothesis, new studies will be necessary. These can comprise studies using computational molecular simulations, further physicochemical experimental studies (e.g., surface plasmon resonance, fluorescence spectroscopy) to determine binding affinities [30] and addressing chitosans with known patterns of acetylation. To the best of our knowledge, this is the first documented evidence that CS might interact with a nucleotide by a mechanism other than non-specific electrostatic binding.

The protection of ASOs and the binding efficiency of ASOs and CS was evaluated with a gel retardation assay. Results revealed that the nucleic acid was retained in the gel pocket by CS with advanced efficiency in higher P/N charge ratios (Figure 4). The findings of the strong binding efficiency of nucleic acids and CS correspond to previous reports [23,33]. Deng et al. even demonstrated protection of plasmid DNA from DNases by complexation with CS and hyaluronic acid [34]. Overall, CS proves to be a suitable carrier for nucleic acids in general.

Finally, nanocomplexes at a P/N charge ratio of 90 prepared in the presence of 85 mM NaCl were chosen for transfection experiments (Table 3). One barrier the nanocomplexes need to overcome in the lung are the respiratory secretions [35]. Atomic force microscopy revealed that the sputum of CF patients displays pores with sizes varying from 300 to 700 nm [36]. Another study reported pore sizes of 100–400 nm discovered by transmission electron microscopy [37]. The same group also evaluated the transport of nanospheres and cationic lipoplexes through a 220 µm thick layer of CF sputum [37]. They report that particles with a size of 120 nm moved through the mucus only with a 1.3-fold retardation compared to an equally sized buffer layer. Particles with a diameter of 270 nm still moved with a 6.8-fold retardation, while the mucus almost completely sterically blocked particles with a diameter of 560 nm. Therefore, with a diameter of 150 nm, the nanocomplexes of the present study are likely to pass through the thick CF mucus. Furthermore, their size makes them promising for transfection purposes, as nanocomplexes smaller than 500 nm are suitable for endocytotic uptake by the cell [38]. The positive zeta potential of approximately +30 mV makes the nanocomplexes not only stable but also able to interact with the negatively charged cell membrane, further facilitating endocytosis [39]. After endocytotic uptake of the nanocomplexes, they are believed to be released from the endosome due to the proton sponge effect hypothesis. In the acidified milieu of the endosome, the NH_2_ groups at the D-glucosamine of CS promote the influx of protons into the vesicle. The transport of counter ions into the endosome to balance the accumulated protons leads to swelling and finally bursting of the vesicle, thus releasing its content [40]. The nucleic acid is then likely released by enzymatic degradation of its carrier. In vertebrates, CS is thought to be primarily degraded by lysozyme or bacterial enzymes in the colon [16]. Moreover, three enzymatically active forms of human chitinases and chitosanases, also known to cleave chitosan, have been identified [41]. One of them, namely the human mammalian chitinase, AMCase, was detected in the lung [42]. Altogether, these studies anticipate that a safe and successful transfection as well as degradation of CS nanocomplexes in human respiratory epithelia is possible.

### 4.2. Supplemented Transfection Medium Stabilizes Nanocomplexes but Harms Human Respiratory Epithelial Cells

To guarantee appropriate biological performance and a successful transfection with the nanocomplexes, it is essential to evaluate their colloidal stability in the transfection medium. In this study, we used Opti-MEM™, as it is commonly used for transfection purposes in various cell lines [43,44]. Furthermore, we supplemented the medium with HEPES and mannitol, based on a recommendation for transfection with the commercially available CS-based transfection reagent, Novafect, from NovaMatrix^®^ (Sandvika, N). Interestingly, HEPES has been reported to be beneficial for transfection purposes before [45,46]. As expected, nanocomplexes were relatively stable in Opti-MEM™ with supplements in contrast to nanocomplexes in the pure medium, even though the size of the nanocomplexes increased slightly (Figure 5). This increase can be explained by interaction of the nanocomplexes with electrolytes and proteins in the medium also observed before [47]. The supplements in the medium likely accumulate in proximity to the hydrophilic surface of the nanocomplexes, producing short-range repulsive hydration forces and thereby stabilizing the nanocomplexes [23,47]. Based on the results of the stability measurements, transfection experiments should be performed with the supplemented medium. However, in order to assure a successful transfection, not only the stability of the nanocomplexes should be taken into consideration, but also the well-being of the cells in general. Therefore, the osmolality of the different cell culture media was assessed and found varying at approximately 270 mOsmol/kg (Table 4). These findings conform with the osmolality of varying cell culture media ranging from 230 to 340 mOsmol/kg [48]. Conversely, the osmolality of Opti-MEM™ supplemented with HEPES and mannitol displayed an osmolality of ~580 mOsmol/kg, making the medium highly hypertonic. This elevated concentration can strongly affect cells, as reported by Kastl et al., who showed that osmolality of 500 mOsmol/kg leads to significant shrinking of pancreatic tumor cells [49]. The negative effect of the supplemented medium was also demonstrated by an MTT assay. The cell viability of NCI-H441 cells was significantly decreased by Opti-MEM™ supplemented with HEPES and mannitol in contrast to the medium without supplements (Figure 6). The increase in cytotoxicity might also be enforced by mannitol, as it has been reported to affect cells negatively [15].

Even though results of the stability measurements suggest that the experiments should be conducted with the supplemented medium, the osmolality of the medium as well as the result of the MTT assay advised otherwise. Hence, transfection experiments were conducted with Opti-MEM™ without supplements.

### 4.3. ASO Transfection Successfully Downregulates ENaC Activity in Human Respiratory Epithelial Cells

There is still a shortage of appropriate cell lines in the field of CF research. For example, the CF bronchial epithelial cell line CFBE41o- lacks significant functional ENaC expression and thus may not be a suitable model system for the study of CF airway pathophysiology [50]. It was reported that CFBE41o- cells do not display an amiloride-sensitive current even after treatment with dexamethasone, suggesting that ENaC stays absent from the cell surface [51]. Furthermore, these cells do not generate a proper mucus when cultivated as ALI cultures [52]. Therefore, even though dysregulated mucociliary clearance is observed more proximally in the CF lung, the distal lung epithelial cell line, NCI-H441, was chosen and cultivated as ALI culture for in vitro transfection experiments, as these cells express sufficient α-ENaC for knockdown purposes [53]. In our study, we focus on the development of a delivery system based on CS in the context of CF. In general, this kind of application could be adaptable to a variety of other issues and is therefore a proof of principle. In the future, it might be interesting to perform transfection experiments and investigate subsequent ion transport via CFTR and ENaC on the cell lines, NuLi (normal lung, University of Iowa) and CuFi (cystic fibrosis, University of Iowa) [54]. These bronchial epithelial cells from healthy and CF donor lungs display the predicted cAMP-dependent as well as amiloride-sensitive currents associated with their genotypes [54]. Finally, the best physiologically relevant samples for the transfection experiments would be primary human nasal epithelial cells acquired from CF patients, as these cells would represent a proper in vivo situation.

To show that successful cellular uptake of ASOs using CS can be achieved, NCI-H441 cells were transfected with the fluorescently labeled 5′Fam-ASOgreen. Figure 7 displays representative CLSM images of transfected cells, showing strong fluorescence after transfection with Lipofectamine as well as CS. These outcomes corroborate the results obtained before proofing the functionality of the fluorescent 5′Fam tag [10]. A similar transfection approach to the one in this study was used by Nafee et al. using CS and poly(lactid-*co*-glycolid) to transfect 2′-*O*-methyl-RNA ASO coupled with a 5′Fam tag into A549 cells, a human lung cancer cell line. CLSM images clearly show the green fluorescence from the 5′Fam inside the cell [22]. These findings prove that CS–ASO nanocomplexes successfully enter human airway epithelial cells within twenty-four hours. Evaluation of total RGB fluorescence intensities showed a highly significant increase in fluorescence after transfection using both Lipofectamine and CS (Figure 8). Even though the transfection efficiency of CS appears to be slightly lower, no significant difference between the two transfection reagents was determined, thus proving CS to be as effective as the commercially available transfection reagent. The results obtained from the MTT assay showing that Lipofectamine significantly decreases the viability of NCI-H441 cells as opposed to CS (Figure 6) are consistent with the notion that transfection with the natural biopolymer is preferable to the lipid-based transfection.

Statistically, a 17 mer oligonucleotide occurs just once in the sequence of the human genome, enabling extremely selective intervention with ASOs of this length [55]. The ASOs used in this study are composed of 16 nucleotides, making the probability of the sequence being selective for α-ENaC extremely high. Therefore, in order to test whether the α-ENaC ASO effectively decreases ENaC activity, NCI-H441 cells were transfected with ASOgreen. Twenty-four hours after transfection, functional Ussing chamber measurements were conducted and the amiloride-sensitive current was evaluated. While transfection with Lipofectamine only led to an insignificant decrease in amiloride-sensitive short-circuit current, the decrease observed after transfection with CS was highly significant (Figure 9c). The difference in transfection efficiency might be explained by the cytotoxic effect of the commercially available transfection reagent, which was revealed by the MTT assay (Figure 6) and was shown previously in other studies [23,56,57]. CS, on the other hand, did not have an effect on the viability of cells, displaying its non-cytotoxicity and potential as transfection reagent. Furthermore, studies revealed that CS has penetration-enhancing effects on cells, increasing its transfection efficiency [58,59]. The findings of decreased ENaC activity after transfection of ASOs directed against the α-subunit of the sodium channel are concordant with previous studies [10,60]. 

The use of ASOs to reduce ENaC activity and thereby evoke rehydration of the airway surface has already been of interest in CF research. Griesenbach et al. could not observe a decrease in ENaC activity after transfection using Genzyme lipid 67 in mice [61], showing that lipid-based transfection is not always suitable. Crosby et al., on the other hand, could successfully decrease ENaC expression and ameliorate inflammation and airway hyperresponsiveness after ENaC ASO inhalation in mice with CF lung disease [62]. However, the pertinence of these findings should be considered with caution, as the murine model of CF often fails to reflect human pulmonary pathology [63]. Nevertheless, these studies demonstrate the importance of targeting ENaC for a successful restoration of ion transport and hence airway surface rehydration in CF.

## 5. Conclusions

The results presented herein corroborate the aspiring opportunities of nanobiotechnology for a possible CF treatment. We designed and characterized CS-based nanosystems complexing three different ASOs, including an ASO directed against the α-subunit of ENaC, demonstrating the ability of the polymer to successfully form small and monodisperse nanocomplexes for transfection purposes. Furthermore, we demonstrated successful uptake of the nanocomplexes by human airway epithelial cells using fluorescence optical methods as well as significant downregulation of ENaC-mediated Na^+^ current after transfection by performing functional transepithelial Ussing chamber measurements. The results presented and discussed in this study demonstrate the promising features of the unique biopolymer chitosan as a non-viral and non-toxic delivery system. 

## Figures and Tables

**Figure 1 biomolecules-10-00553-f001:**
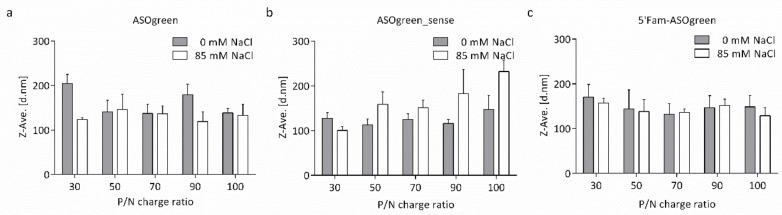
The Z-average hydrodynamic diameter of CS–ASO nanocomplexes at varying P/N charge ratios. The size of CS–ASO nanocomplexes with (**a**) ASOgreen, (**b**) ASOgreen_sense and (**c**) 5′Fam-ASOgreen varied between 100 and 200 nm. On average, nanocomplexes prepared in the presence of 85 mM NaCl were smaller than nanocomplexes prepared without NaCl (*n* = 3).

**Figure 2 biomolecules-10-00553-f002:**
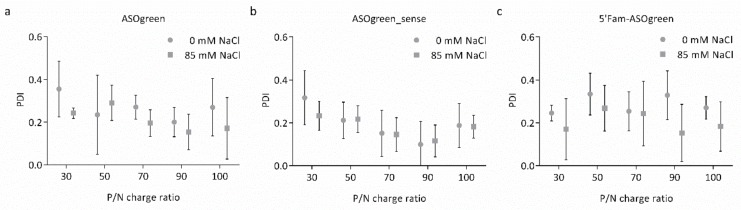
The polydispersity index (PDI) of CS–ASO nanocomplexes at varying P/N charge ratios. The PDI of CS–ASO nanocomplexes with (**a**) ASOgreen, (**b**) ASOgreen_sense and (**c**) 5′Fam-ASOgreen varied between 0.1 and 0.4. On average, nanocomplexes prepared in the presence of 85 mM NaCl had a lower PDI than nanocomplexes prepared without NaCl. The lowest PDI was observed in nanocomplexes at a P/N charge ratio of 90 (*n* = 3).

**Figure 3 biomolecules-10-00553-f003:**
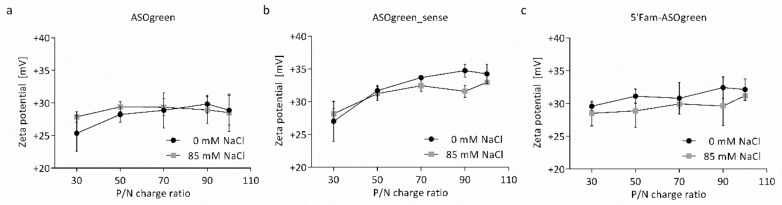
The zeta potential of CS–ASO nanocomplexes at varying P/N charge ratios. The zeta potential of CS–ASO nanocomplexes with (**a**) ASOgreen, (**b**) ASOgreen_sense and (**c**) 5′Fam-ASOgreen increased with increasing P/N charge ratio. On average, nanocomplexes prepared in the presence of 85 mM NaCl had a lower zeta potential than nanocomplexes prepared without NaCl (*n* = 3).

**Figure 4 biomolecules-10-00553-f004:**
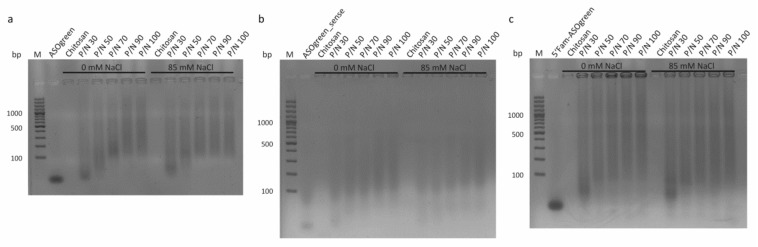
Gel retardation assay of CS–ASO nanocomplexes at varying P/N charge ratios. The result of a 1.5% agarose gel electrophoresis of CS–ASO nanocomplexes with (**a**) ASOgreen, (**b**) ASOgreen_sense and (**c**) 5′Fam-ASOgreen prepared with and without 85 mM NaCl is shown. Naked ASOs showed a band below 100 bp. CS was retained in the pocket. The retained amount of ASOs by CS was proportional to the charge ratio. Marker (M): O’Gene Ruler 100 bp Plus (Thermo Fisher Scientific, Waltham, MA, USA).

**Figure 5 biomolecules-10-00553-f005:**
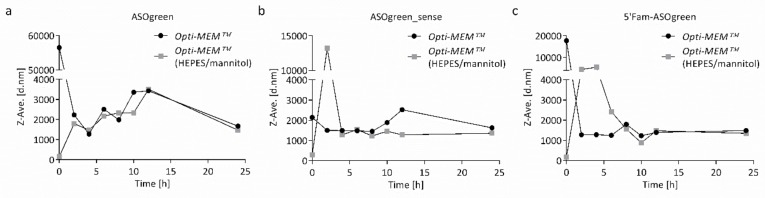
Stability of CS–ASO nanocomplexes in the transfection medium. The stability of CS–ASO nanocomplexes with (**a**) ASOgreen, (**b**) ASOgreen_sense and (**c**) 5′Fam-ASOgreen at a P/N charge ratio of 90 with 85 mM NaCl is shown. Nanocomplexes were incubated in Opti-MEM™ or Opti-MEM™ supplemented with HEPES (20 mM) and mannitol (270 mM) at 37 °C. Nanocomplexes in Opti-MEM™ supplemented with HEPES and mannitol appeared to be more stable than nanocomplexes in Opti-MEM™ alone.

**Figure 6 biomolecules-10-00553-f006:**
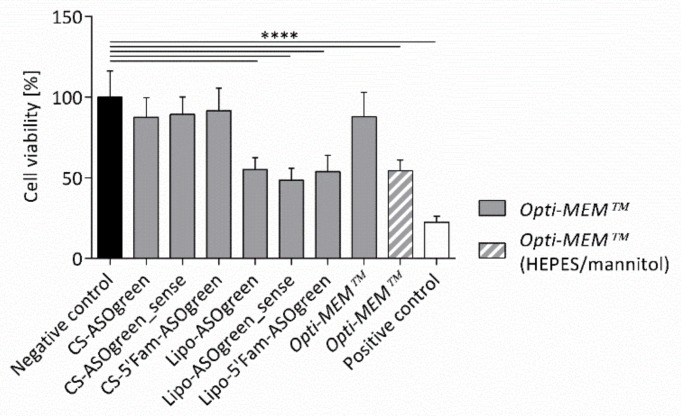
Effect of different ASOs in combination with Lipofectamine (Lipo) and CS on the viability of NCI-H441 cells. NCI-H441 cells were incubated with the samples for 24 h before a 3-(4,5-dimethylthiazol-2-yl)-2,5-diphenyltetrazolium bromide (MTT) assay was conducted. Cell culture medium was used as a negative control; Triton^®^ X-100 was used as a positive control. While CS in combination with ASOs only had a slight effect on the cell viability (~80–90%), the decrease in cell viability caused by Lipofectamine in combination with ASOs was highly significant (~50%; *p* ≤ 0.0001 (****); *n* = 3).

**Figure 7 biomolecules-10-00553-f007:**
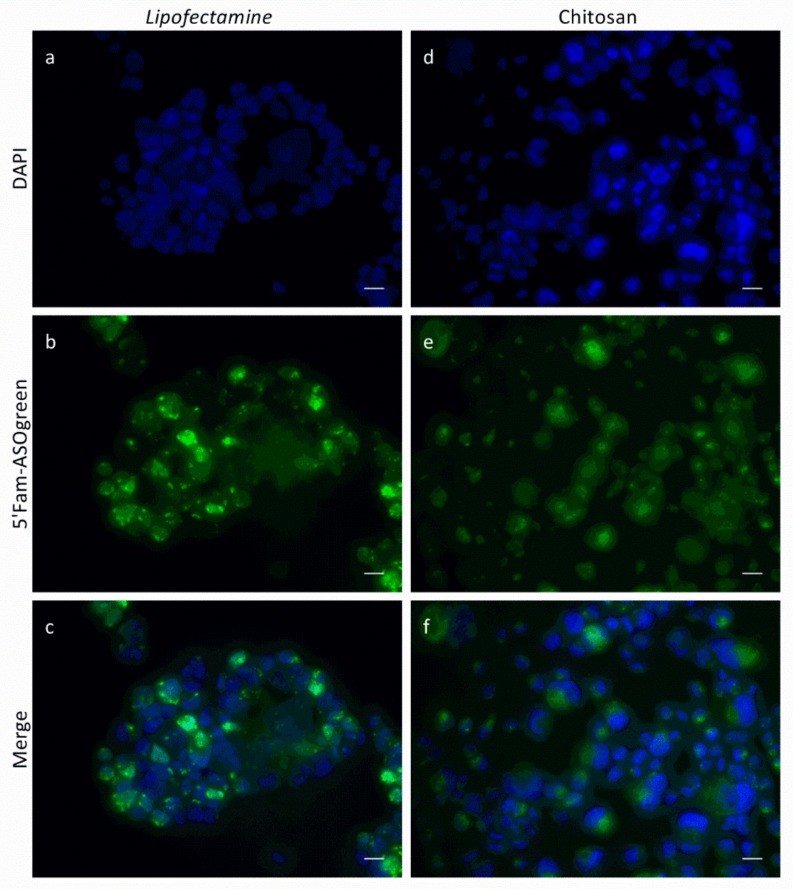
Representative confocal laser scanning microscope (CLSM) images of NCI-H441 cells transfected with 5′Fam-ASOgreen. Cells were transfected using (**a**–**c**) Lipofectamine and (**d**–**f**) CS. Images were taken 24 h after transfection. (**a**,**d**) DAPI; (**b**,**e**) 5′Fam-ASOgreen; (**c**,**f**) Merge (scale bar = 20 µm).

**Figure 8 biomolecules-10-00553-f008:**
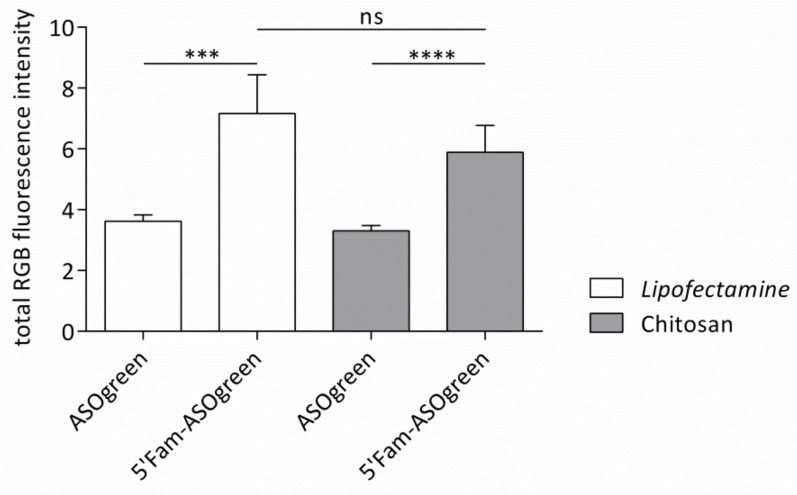
Fluorescence intensities of NCI-H441 cells transfected with ASOgreen and 5′Fam-ASOgreen. Cells were transfected with the fluorescent 5′Fam-ASOgreen as well as the non-fluorescent ASOgreen as controls. Lipofectamine and CS were used as transfection reagents. Total RGB fluorescence intensity was determined 24 h after transfection (non-significant (ns), *p* ≤ 0.001 (***), *p* ≤ 0.0001 (****); *n* = 12).

**Figure 9 biomolecules-10-00553-f009:**
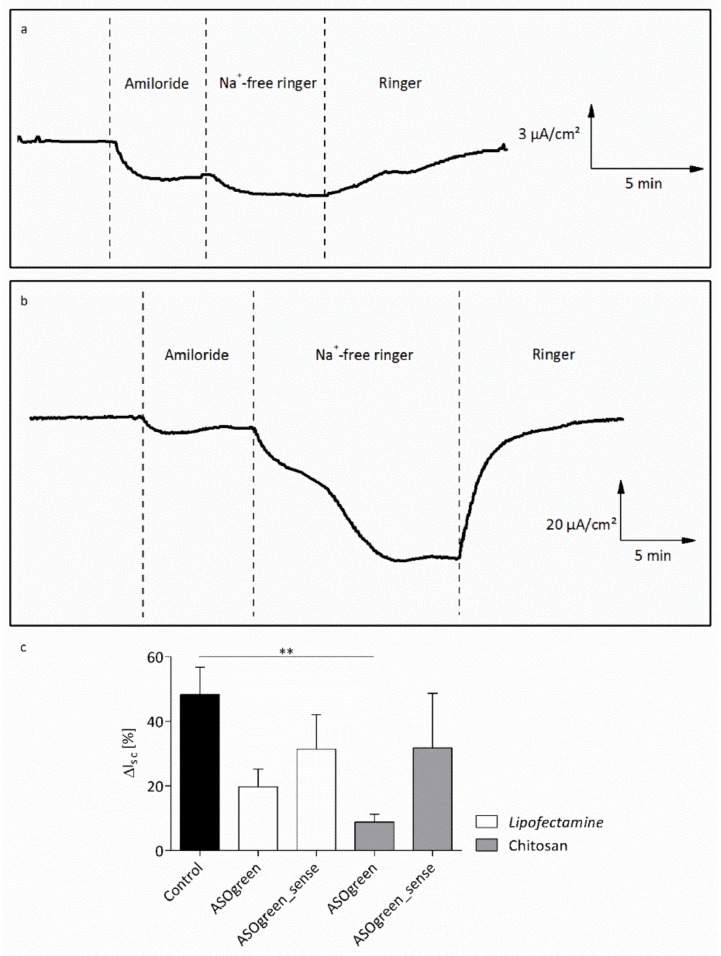
Transepithelial Ussing chamber measurements of NCI-H441 cells. (**a**) Representative time course of non-transfected cells, showing a slight decrease in short-circuit current (I_sc_) after application of Na^+^-free ringer solution. (**b**) Representative time course of cells transfected with ASOgreen using CS (0.45 μg/cm² ASOgreen), showing a strongly decreased I_sc_ after withdrawal of Na^+^. Please note the different scales for I_sc_. (**c**) Statistical evaluation of amiloride-sensitive short-circuit current (ΔI_sc_) in NCI-H441 cells after ASO transfection with Lipofectamine and CS. Cells were transfected with 0.45 μg/cm² ASO, respectively; control cells were not transfected. Measurements were conducted 24 h after transfection. The amiloride-sensitive current decreased after transfection with ASOgreen. Cells transfected with ASOgreen_sense only showed a slight decrease in the amiloride-sensitive current (*p* ≤ 0.01 (**); control, *n* = 8; ASOgreen, *n* = 7; ASOgreen_sense, *n* = 5).

**Table 1 biomolecules-10-00553-t001:** α-epithelial sodium channel (ENaC) antisense oligonucleotides.

Oligo Name	Sequence 5′-3′	Orientation
ASOgreen	TGG ATG GTG GTG TTG T	antisense
ASOgreen_sense	ACA ACA CCA CCA TCC A	sense (negative control)
5′Fam-ASOgreen	6-Fam-TGG ATG GTG GTG TTG T	antisense (fluorescent)

**Table 2 biomolecules-10-00553-t002:** Composition of chitosan (CS)–antisense oligonucleotide (ASO) nanocomplexes at varying positive/negative (P/N) charge ratios.

Oligo Name	Charge Ratio	ASO		Chitosan	
	P/N ^1^	(nmol) ^2^	(µg/µL)	(nmol) ^3^	(µg/µL)
	30	4.6	0.3	137.0	2.8
	50	4.6	0.3	228.4	4.6
ASOgreen	70	4.6	0.3	319.7	6.5
	90	4.6	0.3	411.0	8.3
	100	4.6	0.3	456.7	9.3
	30	4.8	0.3	144.3	2.9
	50	4.8	0.3	240.5	4.9
ASOgreen_sense	70	4.8	0.3	336.7	6.8
	90	4.8	0.3	433.0	8.8
	100	4.8	0.3	481.1	9.8
	30	4.7	0.3	140.0	2.8
	50	4.7	0.3	233.1	4.7
5′Fam-ASOgreen	70	4.7	0.3	326.4	6.6
	90	4.7	0.3	419.6	8.5
	100	4.7	0.3	466.2	9.5

^1^ Charge ratio (P/N): molar ratio of equivalent charges of NH_3_^+^/PO_4_^−^, ^2^ ASO (nmol): equivalent concentration of PO_4_^−^ from the ASO. ^3^ Chitosan (nmol): equivalent concentration of NH_3_^+^ from CS.

**Table 3 biomolecules-10-00553-t003:** Physicochemical properties of CS–ASO prototype nanocomplexes at a P/N charge ratio of 90 prepared with 85 mM NaCl.

Attribute	ASOgreen	ASOgreen_sense	5‘Fam-ASOgreen
Hydrodynamic diameter [d.nm]	119.0 ± 17.9	182.9 ± 44.3	151.7 ± 11.6
PDI	0.15 ± 0.07	0.12 ± 0.06	0.15 ± 0.11
Zeta potential [mV]	+28.9 ± 1.7	+31.6 ± 0.7	+29.6 ± 2.4

**Table 4 biomolecules-10-00553-t004:** The osmolality of cell culture and transfection media.

Medium	Osmolality (mOsmol/kg)
NCI-H441 cell culture medium	270.3 ± 1.7
Opti-MEM™	272.7 ± 2.6
Opti-MEM™ + HEPES (20 mM) + mannitol (270 mM)	582.7 ± 4.0

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
