# Peer review of "Chitosan Nanocomplexes for the Delivery of ENaC Antisense Oligonucleotides to Airway Epithelial Cells"

_biomolecules, 2020, doi:10.3390/biom10040553_

Round 1
Reviewer 1 Report
See the attachment

Reviewer 2 Report
The manuscript entitled “Chitosan nanocomplexes for the delivery of ENaC antisense oligonucleotides to airway epithelial cells” by A. Katharina Kolonko, Nadine Bangel-Ruland, Francisco M. Goycoolea and Wolf-Michael Weber reports development of a chitosan-based antisense oligonucleotide against ENaC for cystic fibrosis (CF) treatment.
Although CF is a devastating disease and the approach of using antisense oligonucleotides is very interesting, the authors have chosen to test their agent in H441 cells which are originally derived from a papillary adenocarcinoma of the lung and are rather used as a model for cell types of the distal lung. However, the pathophysiology of CF is strongly dependent on the dysregulated mucociliary clearance observed more proximally in the lung, as the authors nicely mention in the manuscript. I believe it would be beneficial to mention this in their discussion with possible measures which can be taken in future studies.
Round 2
Reviewer 1 Report
Accept